# Criteria for Assessing Exposure to Biomechanical Risk Factors: A Research-to-Practice Guide—Part 2: Upper Limbs

**DOI:** 10.3390/life15010109

**Published:** 2025-01-16

**Authors:** Francesca Graziosi, Roberta Bonfiglioli, Francesco Decataldo, Francesco Saverio Violante

**Affiliations:** 1Occupational Medicine Unit, Department of Medical and Surgical Sciences, Alma Mater Studiorum University of Bologna, 40138 Bologna, Italy; francesca.graziosi@unibo.it (F.G.); roberta.bonfiglioli@unibo.it (R.B.); francesco.violante@unibo.it (F.S.V.); 2Division of Occupational Medicine, IRCCS Azienda Ospedaliero-Universitaria di Bologna, 40138 Bologna, Italy

**Keywords:** biomechanical risk factors, occupational diseases, musculoskeletal disorders, ergonomics, occupational medicine

## Abstract

Musculoskeletal disorders are the most prevalent occupational health problem and are often related to biomechanical risk factors. Over the last forty years, observational methods for exposure assessment have been proposed. To apply them effectively in the field, an in-depth knowledge of each methodology and a solid understanding of their actual predictive value and limitations are required. In this two-part guide, we discuss methods that have a solid scientific background, are based on expert consensus, and that do not require disproportionate technical, material, financial, and time resources. In Part 1, we focused on the Revised NIOSH Lifting Equation as a validated method for assessing manual material handling and discussed its application when dealing with task variability. In Part 2, we look at methods for the assessment of upper-limb biomechanical exposure in manual jobs. According to the above-mentioned criteria, we discuss methodologies proposed by the American Conference of Governmental Industrial Hygienists (ACGIH) and evaluate activities requiring high-speed continuous movement and the use of hand force, working with the arms above the shoulder level, to prevent localized fatigue in the upper extremities in cyclical work tasks. Finally, a preliminary proposal of a proportionate risk assessment of working duration in part-time jobs is presented.

## 1. Introduction

In the first of this two-part guide, general issues regarding occupational biomechanical risk assessment were discussed, together with a (consensus-based) commentary of the most documented (and used) methods for the risk assessment of occupational manual material handling [1,2].

In the last 40 years, much work has been performed to characterize the biomechanical risk associated with manual work involving force, speed, and continuity of movement [3], which has emerged as a relevant issue, on par with manual material handling and awkward posture [4]. Several disorders of the upper limbs, linked with variable degrees of evidence to manual force, velocity, and repetitive movements [5], are still prevalent nowadays [6]; thus, it is extremely important to ensure that workers are not exposed to conditions which may increase the likelihood of developing musculoskeletal upper-limb disorders, which are frequent, anyway, among the general population. According to a systematic review, over the years, different methodologies have been proposed in the scientific literature to assess occupational exposure to biomechanical risk factors [7]. Some methods can be considered “general”, i.e., such that they provide an assessment of whole-body biomechanical overload to all risk factors (e.g., the Washington State Checklists https://lni.wa.gov/safety-health/_docs/CautionZoneJobsChecklist.pdf, accessed on 3 September 2024 and https://lni.wa.gov/safety-health/_docs/HazardZoneChecklist.pdf, accessed on 3 September 2024). Other tools analyze more in-depth specific biomechanical parameters (e.g., the RULA method assesses postural loading in more detail) [8].

In some occupational settings where workers perform daily manual activities and repeated exertions, muscle fatigue can also occur. Muscle fatigue can be defined as “any reduction in the ability to exert force in response to a voluntary effort” [9]. Fatigue also induces discomfort and pain, and in the long term can contribute to the development of connective tissue disorders. For this reason, it is important to predict and limit fatigue in order to preserve workers’ strength, health, and work performance. Localized muscle fatigue can either be assessed objectively (i.e., surface electromyography), or by rating perceived exertion (RPE) [10].

Despite an abundance of methods, in the absence of a “gold standard”, practitioners need recommendations to approach exposure evaluation on- field. In this work, we present and discuss methods for the biomechanical risk assessment of the upper limbs with a solid scientific background and expert consensus. These methods do not require disproportionate technical, material, financial, or time resources.

In addition, we suggest an approach to “translate” the results of the biomechanical risk assessment, often in the form of numerical values/indices, into classification bands to orient ergonomists, occupational health and safety specialists, employers, and worker representatives to identify priorities and set up preventive measures. Our aim is to bridge the gap between research studies and practice, and also to provide practical recommendations.

Finally, we discuss how to apply the biomechanical risk assessment to working duration in part-time jobs, which are now widely represented in several sectors [11,12], potentially scaling down the risk level.

## 2. Methods Selected for the In-Depth, Second-Level Analysis of Manual Activities

In this section, we present methodologies that can be used for an in-depth evaluation (second-level analysis) of manual activities after the preliminary screening (first-level analysis) described in Part 1 of this guide [2]. The selection process was based on both a literature review of international scientific papers and expert consensus. Methods with a solid scientific background that have been published in peer-reviewed journals or recognized books in the occupational field, and with a clear description written in the English language, are chosen. We describe and discuss one method to assess biomechanical overload in the distal part of the upper limb (Section 2.1), one tool for the evaluation of work performed above the shoulder level (Section 2.2), and a procedure to define the acceptability of a task in terms of upper-limb fatigue (Section 2.3).

### 2.1. Manual Work Requiring Speed, Continuity of Movement, and Use of Force

For the second-level analysis of manual activities requiring speed, continuity of movement, and use of force, the method for which more validation studies are available is the one proposed by the ACGIH (formerly the American Conference of Governmental Industrial Hygienists) [13].

The ACGIH method is focused on the evaluation of job risk factors associated with musculoskeletal disorders of the hand, wrist, and forearm, and is derived from epidemiological, psychophysical, and biomechanical studies. It is applied to work activities that involve a series of repetitive and similar actions or movements carried out from 4 to 8 h a day, and it is aimed at reducing musculoskeletal disorders of the hand by configuring conditions to which almost all workers can be exposed without reporting any damage to health. For practical purposes, if the overall exposure of workers to repetitive tasks is less than 4 h, biomechanical analysis can still be carried out according to the ACGIH criteria, but the results obtained would lead to an overestimation of risk. On the other hand, if the overall duration of the repetitive task is less than 1 h per day or 5 h per week, even the ISO 11228-3 standard [14] considers the “repetitiveness” risk factor as negligible; therefore, there would be no need to carry out a specific assessment [14].

In 1999, the authors of the method published the first cross-sectional study assessing the relationship between exposure to repetitive tasks and the prevalence of musculoskeletal diseases of the upper limb; workers assigned to repetitive tasks presented a risk of developing musculoskeletal diseases of the upper limb (tendinitis, carpal tunnel syndrome) that increased proportionally to the level of repetitiveness itself [15].

The method underwent a validation process through many cross-sectional [16,17,18] and longitudinal [19,20] studies conducted in different settings, such as the manufacturing, healthcare, clerical, and engineering sectors. Recently, longitudinal studies in manufacturing have confirmed the ability of the ACGIH TLV to predict the onset of carpal tunnel syndrome and tendonitis [21,22,23,24], and it continues to be reviewed and used as a gold standard [25,26].

Inter-observer and intra-observer repeatability were also investigated and judged as moderate to good [27,28]; the method showed a moderate-to-good correspondence with the results of other tools, such as the Strain Index and video analysis [18,21,29,30].

The method is included in the ISO standard 11228-3 [14] for the in-depth analysis (second level) of repetitive tasks involving the handling of light loads at high frequency.

The ACGIH method identifies a threshold limit value (TLV) through the combination of 2 parameters: the hand activity level and the Normalized Peak Force, which, as described by the method itself (ref. [13]) can be determined as follows:

Hand Activity Level (HAL)

HAL can be determined by evaluating the average frequency of hand movements during a work cycle and the duration of the “duty cycle” (i.e., the distribution of actual work, where manual effort is greater than 10% of the force exerted in the specific posture, and recovery/rest time).

Alternatively (or in addition), HAL can be estimated by a trained observer who scores the manual task based on an analog scale from 0 (no repetition, hands idle most of the time) to 10 (rapid movements, continuous exertions).

Normalized Peak Force (NPF)

Peak manual force is the maximum force exerted by the hand during each usual work cycle. The peak force is expressed as a value from 0 to 10, which corresponds to the percentage of force used in the task compared to the reference force applicable by the general population for performing the same task: the value is therefore “normalized” on a scale from 0% to 100%. The peak force can be obtained by the following:-The observation of a trained operator;-The judgment of the worker involved in the manual operation using a subjective perception scale, such as the Borg scale;-The use of force-measuring instruments;-Surface electromyographic techniques;-The use of biomechanical equipment or models.

The combination of HAL and NPF results in a specific level of exposure for the task examined, which must be compared with the TLV proposed by the ACGIH: if the TLV is exceeded, the working conditions result in a significantly increased incidence of hand musculoskeletal disorders.

To ensure greater conditions of protection (it is conceivable that groups of workers are less resistant to carrying out operations that require repetitive movements and/or the use of force), the ACGIH has also introduced an Action Limit (AL), which represents an additional guard level. If occupational exposure is in the area between the TLV and AL, the ACGIH recommends that general control measures should be taken.

Alternatively, as reported in the ACGIH manual, the Peak Force Index (*PFI*) can be calculated as follows:PFITLV=NPFNPFTLVPFIAL=NPFNPFAL
where *NPF_TLV_* and *NPF_AL_* are the peak force values resulting from the intersection of the HAL score (observed) with the TLV and AL lines, respectively [26]. For a PFI greater than 1, the respective limit is exceeded.

In multi-tasks jobs, the evaluation can be carried out as a time-weighted average (TWA) according to three different modalities, as reported in the ACGIH manual.

### 2.2. Work with the Arms Above the Shoulders

Threshold limit values (TLVs) for high-intensity work with the hands at or above shoulder height were recently (2023) recommended by the ACGIH [13]. Given their recent introduction, no field studies are available for these TLVs. However, they are based on a solid scientific background and, being specific for assessing manual tasks performed above shoulder height, they may be useful for designing acceptable working conditions [31]. It is thought that most workers can be exposed, day in and day out, below these thresholds without experiencing work-related weariness or shoulder issues. Specifically, the values reported are protective for 75% of women and 95% of men.

The hand’s position relative to the shoulder when the hand exerts force upward or forward is the basis for TLVs. Three axes are used to measure how the hand is positioned in relation to the shoulder: vertical (upper), anterior (forward), and medial–lateral (lateral). The acromion is used as measurement point on the shoulder (bony reference point on the upper front of the shoulder); the position of the hand measurement point is in the center of the palm, at the carpometacarpal joint of the third finger. It is assumed that the wrist has a neutral posture.

The TLVs for a single-handed forward (upward) push force (in kilograms) are defined for various medial–lateral hand locations relative to the shoulder, highlighting the maximum pushing force that might be exerted.

When the force exerted by the hand exceeds the threshold limit of force (TLV), it is recommended to employ suitable control measures to mitigate the risk of shoulder disorders. These measures may include lowering hand height, decreasing hands’ reach forward, shortening hands’ force application duration, or requiring less force to accomplish the task. It is worth mentioning that regular over-the-shoulder work, especially if requiring significant force, can also cause shoulder fatigue, increasing the risk of discomfort. Thus, to prevent fatigue in repetitive operations, the ACGIH proposes a specific approach, described in the following Section 2.3.

Finally, other manual material handling operations that might result in shoulder strain or injury, such as pushing/pulling or lifting/carrying loads below shoulder height, may coexist with over-the-shoulder work (for example, in agricultural operations or maintenance activities). If this is the case, a further lowering of the developed force might be required; threshold values for handling activities performed under the shoulder level may be found in the literature [32,33].

### 2.3. Localized Fatigue in the Upper Extremities

The ACGIH has recommended a TLV to prevent upper-limb fatigue in cyclical work tasks, which has recently (2022) been integrated with further additions [13]. It is thought that most healthy workers can be exposed to conditions under the TLV acceptable level daily. If this is the case, they are expected to maintain their working capacity and normal performance, and they should not experience extreme or continuous musculoskeletal fatigue across the whole upper limbs (hands/wrists, forearms, elbows, and shoulders).

The TLV is mainly based on psychophysical data [34] and has been applied in both field and laboratory studies [35,36].

The issue of localized fatigue, including several elements, mechanisms, and consequences resulting from physical exertion, is not trivial. It impacts the musculoskeletal system’s comfort and capacity to carry out everyday tasks, work, and leisure activities. Localized discomfort, pain, tremors, diminished strength, and other symptoms or indicators of impaired motor control can be evidence of fatigue. In the context of the ACGIH TLV, fatigue is defined as discomfort or diminished upper-limb function that manifests itself within 24 h following extended or recurrent strain of the hands and arms. Any indication or symptom that lasts longer than 24 h should be considered as a potential musculoskeletal problem related to the workplace. Weariness may be a sign of long-term soft tissue damage.

By itself, a certain level of localized weariness is harmless. Fatigue is a natural physiological response and an aspect of life, and it can even help musculoskeletal tissues adjust to physical stress and/or unfamiliar activity. However, it should not last from one working day to the next or affect everyday activities or work-related tasks. Furthermore, workers may need a few days or even weeks to psychologically and physically acclimate to a new job or task. Throughout this adaptation phase, abnormal symptoms could appear. Localized fatigue occurring throughout the workday should be reversed during daily breaks from work, allowing the worker to complete regular work functions and after-work activities.

Workload Patterns

According to the ACGIH TLV, workload pattern refers to the ability to repeat and/or sustain biomechanical loads over time. This means that forces and force moments maintain a regular temporal pattern to perform a task (i.e., the same effort required to maintain the worker’s weight and the weight of tools, as well as to use them as needed to complete a task).

By dividing the applied moments or forces by the strength of the relevant joint and by the posture of an individual or population of interest, loads can be normalized by force. The greatest force or moment that the body segment of interest is willing to produce is referred to as strength. Normalized loads can generally be represented as a percentage from 0% to 100%, or on a scale from 0 to 10. Another common way to express normalized loads is as a percentage of the maximum voluntary contraction (%MVC).

A variety of methods can be used to estimate loads, including direct measurements (such as dynamometer), indirect measurements (such as electromyography), worker-perceived stressors, biomechanical calculations, and observations by highly trained experts. Worker’s strength can be assessed directly or indirectly (using biomechanical models or population research). The type of working activity and the worker’s characteristics will determine the optimal strategy. Procedures for studying the work-induced load are available in the literature.

A fatigue curve of the TLV is proposed by the ACGIH, expressed by the equation of %MVC in the function of the duty cycle:(1)$$\%MVC=100\%\cdot \left(-0.143\cdot \ln\left(\frac{DC}{100\%}\right)+0.066\right)$$
where %MVC is the percentage of maximum force or effort of the hand, elbow, or shoulder, and DC is the duty cycle, expressed as the percentage of time during which force is applied compared to the whole activity. Using this fatigue curve/equation, an acceptable duty cycle for a given force (%MVC) or an acceptable %MVC for a certain duty cycle can be mathematically determined.

The TLV is meant for cyclical work that is typically completed in two or more hours each day; if a worker completes many tasks that need two hours or more each, then none of the tasks can surpass the TLV. It is best to limit static hand, elbow, or shoulder efforts to less than 20 min.

In 2022, ACGIH also made explicit how to calculate the minimum recovery time (RT) from the duty cycle (DC) equation (DC = ET/(ET + RT)):RT = (ET/DC) − ET
where ET is the effort time and DC can be derived from the inverse of Equation 0 as a function of MVC.

The equation can be used on the applicable range of the TLV, i.e., 0.5% to 90% DC, which corresponds to stress levels ranging from 10% to 80% MVC, approximately.

## 3. Correspondence Between Numerical Indexes of Biomechanical Load and Other Workload and Risk Assessment Classification Models

The calculation of the numerical indices representing biomechanical load is an important input for risk management in the workplace, hence the need to “translate” these numerical indices into workload classification bands (useful at individual worker level) and risk classification bands (for preventive measures).

For the classification of work activities into bands of overall biomechanical load, the system considered most valid at international level is described in the United States Dictionary of Occupational Titles (DOT) of the United States Department of Labor (now O’NET—Occupational Information Network—accessible at https://occupationalinfo.org/appendxc_1.html#STRENGTH, accessed on 3 September 2024) and has been taken up in the international reference text for fitness for work evaluation [37]. It classifies work activities, in terms of maximum oxygen consumption and manual handling of loads, into five categories.

For the classification of work activities (with the aim of adopting prevention measures), the de facto standard at international level is described in the British Standard 8800 [38], which is the reference for all companies with a certified occupational safety management system with respect to the ISO 45001 standard; in this (voluntary) standard, occupational risk is classified into five bands, to which different preventive actions correspond.

On the other hand, the biomechanical load classification models previously described (NIOSH, HAL ACGIH) classify work activities into three categories (NIOSH: Lifting Index (LI) up to 1, LI between 1 and 3, and LI greater than 3; HAL ACGIH: less than AL, between AL and TLV, and greater than the TLV); hence, there is a need to find a correspondence model between numerical indices, the classification of the Dictionary of Occupational Titles, and the risk category BS 8800.

### 3.1. Manual Material Handling of Vertical Loads (MMHv): Correspondence Between Lift Index (LI), DOT Classification, and BS 8800

A proposed match between maximum typical lifting indices and the DOT classification of work tasks as an operational tool to adapt numerical indices to individual workload evaluation is reported in Table 1 below. To be precautionary, the division of the lifting indices into bands is not proportional.

Table 2 shows the correspondence between lifting indices and the BS 8800 classification of occupational risk as an operational tool to adapt numerical indices to the needs of risk management. To be precautionary, the division of the lifting indices into bands is not proportional. Similarly, other standards and the scientific literature suggest different levels of classification for interpreting Lifting Index (LI) values and recommend priority actions [39,40].

### 3.2. Manual Work Requiring Speed and Continuity of Movement and Use of Force: Correspondence Adopted Between Manual Activity Level, DOT Classification, and BS 8800

To adapt the ACGIH model, which provides a classification into three categories, to the needs of workload evaluation and risk assessment, requiring five classification bands, two additional lines (labeled A and B) are defined (Equations (2) and (3)) by dividing the area between the original AL and TLV lines (Equations (1) and (4), reported in the ACGIH manual [41] and in [26]) homogeneously, as shown in Figure 1.

The equations used are as follows:AL line:   NPF_AL_ = 3.6 − 0.56 × HAL(2)A line:    NPF_A_ = 4.26 − 0.56 × HAL(3)B line:    NPF_B_ = 4.92 − 0.56 × HAL(4)TLV line:  NPF_TLV_ = 5.6 − 0.56 × HAL(5)

By substituting the estimated HAL value in the line equations above, it is possible to obtain the NPF value corresponding to each line and compare it with the peak force value rated for the task (NPF_ESP_). For example, if NPF_ESP_ is between NPF_A_ and NPF_B_, the task is classified as Medium (Medium Risk according to the BS8800 classification).

A proposed match between the ACGIH model and the DOT (for homogeneity, having the same DOT terminology) is reported in Table 3 below.

Table 4 shows the correspondence between the five proposed categories in which the ACGIH model has been classified and the different actions to manage the risk (BS8800 classification).

## 4. Postures

A sufficient literature consensus states the effects of awkward postures assumed and maintained for a long time during the performance of work activities on the development of musculoskeletal diseases affecting different body districts. For each joint, there is an optimal area within which an effort can be made with minimum fatigue and acceptable tissue load.

Traditionally, postures are divided into static (held for a certain time and without support) and dynamic (which are assumed for a limited time, usually during the movement of a part of the body). Until recently, no threshold limit values have ever been set for posture factors; however, reference documents such as the ISO 11226 standard [42] (for unsupported postures maintained for at least 4 s) and texts from the scientific literature (for example, ref. [43]) were available.

The evaluation of static postures can usually be based on the simple observation (possibly assisted by photos or video recordings) of the parts of the body involved (position of the entire body, neck and trunk, upper limbs, lower limbs) and the relative maintenance times. Using the reference values indicated by the scientific literature, it is possible to assess whether the posture assumed for the specific body segment is acceptable or not recommended.

For dynamic postures, judgment is usually integrated into the evaluation of the mode of execution of a specific task (e.g., trunk posture when lifting a load or wrist posture when performing a hand movement).

### Static Postures of the Lower Limbs

Few epidemiological data exist on lower-limb (hip and knee) disorders attributable to work; for preventive purposes, however, it is useful to limit the time in which a worker has to kneel or squat without support. If kneeling work has to last for few minutes, it is necessary to equip the worker with padded mats or knee pads (which should always be used when working on knees).

Another issue might be the need to frequently bend the knees, keeping the foot on the ground (such as when, for example, one wants to lower oneself while keeping the back straight). In this case, the scientific literature does not provide indications of risk; however, it should be considered that this movement involves the lowering and subsequent raising of the trunk for the lower limb which, in terms of weight, corresponds to about two-thirds of the entire body.

The limiting factors are, mainly, localized fatigue of the lower limbs and total energy consumption (a limit that, from a practical point of view, is reached after localized fatigue). In women, localized lower-limb fatigue (defined as a reduction in maximum voluntary contraction capacity of 25%) was experimentally induced in healthy volunteers with 50 alternate knee extensions and flexions at 50% of the maximum voluntary contraction capacity [44]. Another study has shown that between men and women, the differences in terms of ability to perform these movements are similar [45].

Based on these data, it is reasonable to assume that activities that repeatedly involve the need to crouch down and get up (without handling loads) should be limited to 50 movements per hour. This does not apply to lifting or lowering loads while flexing the knees, as this evaluation is included in the NIOSH model (physiological criterion).

## 5. Calculation of Work Task Duration

In the case of tasks that vary in duration and frequency, since there must necessarily be an “average” index, a rational way to proceed is to calculate the average weekly duration for each task by taking as a reference the total duration of the task on an annual basis and dividing it by the weeks worked in the year.

For example, if a given task lasts 920 h in a year, the average weekly duration of the task will be 20 h, referring to 46 weeks worked in the year.

This calculation model makes it possible to obtain an average duration of performance of a given task, regardless of the contract of the worker; therefore, it can be used both for those who work full time and for those who work on any type of part-time arrangement.

## 6. Risk Assessment Weighting According to Exposure Duration

Most scientific studies on work-related occupational disorders are carried out in highly standardized industrial environments, where the tasks of most workers are repeated identically day after day and where working time is regular as well.

In other work areas, however, there is a high variability of tasks within a single working day, as well as variable shift duration (full time, part time in all possible forms).

Since any rational assessment of risk depends on exposure, measured in terms of magnitude, frequency (how often), and duration (how long in total), once the magnitude of any exposure has been evaluated, it is necessary to evaluate its frequency and duration.

Empirical evidence shows that those working part time have a very low (or zero) risk of biomechanical-factor-induced diseases compared to those who work full time [46,47,48,49].

This basic principle of occupational medicine is so general that accidents (occupational injuries) do not escape this rule; in fact, even the frequency of these events in those who work part time is reduced compared to those who work full time [50].

A recent study [51] built a mathematical relationship capable of proportioning the evaluation of biomechanical load as a function of the hours worked per week, based on studies that have analyzed the duration of work, normalizing the risk of those who work full time to 100% and reducing the risk of those who work part time to the hours worked as a percentage. The correlation (with R^2^ > 0.6) between hours worked per week and risk highlights an approximately linear trend of risk reduction upon the working hours decrement (2.6% per hour). For example, a person working 20 h/week is exposed to a biomechanical risk that is approximately halved compared to workers carrying out the same activities for 40 h/week.

To transpose these experimental data to the practice of biomechanical risk assessment, the following must be considered:-Small reductions in weekly hours compared to the maximum (40 h) are unlikely to significantly reduce the risk;-Below a few hours per week, the risk is probably completely absent, due to the large recovery time available, and therefore the starting risk category (very low) must be attributed to a time above that threshold.

This provides the possibility of classifying a given task, from a biomechanical point of view, according to the hours worked; regarding the previously reported criteria for biomechanical risk evaluation, they can be proportionate to the actual hours worked in a task.

When assessing the reduction in risk as a percentage of hours worked, it should be considered that a small reduction in hours (e.g., 4 h, equal to a reduction of 10%) is not likely to have a significant influence, and therefore should be neglected. Larger reductions in working hours (e.g., 8 h, corresponding to a 20% reduction in risk) can instead be considered significant, allowing movement into the next risk category.

For a task that results, for example, in a manual labor risk category of “High” for a worker performing it for 40 h, it will be assumed that the risk remains the same down to 36 h and then lowers by one level (from “High” to “Medium”) if the task is performed for 35 to 27 h; moreover, the risk lowers another level (from “Medium” to “Low”) if the task is performed for 26–18 h, and again (from “Low” to “Very Low”) if the task is performed for less than 18 h.

## 7. Limitations and Future Directions

One of the main limitations affecting the methods for the risk assessment of manual jobs is related to the estimation of force. The simultaneous contribution of internal (muscle contraction) and external (the load handled) components poses challenges in objectively evaluating or measuring the forces actually applied by workers and the loads applied on tissues. The use of wearable technologies, which measure pressure/force (kinetics) and upper-arm posture in real time while assessing kinematic muscle fatigue, may help to improve this drawback.

However, it has to be noted that the introduction of wearable sensors to monitor manual work in field applications could be complicated, requiring economic resources and signal processing skills after raw data acquisition. Furthermore, sensors’ intrusiveness could hamper the feasibility of data recording in real workplaces scenarios (such as in healthcare settings).

Lab-based simulated case studies could be of help, providing benchmark data to compare with in-field biomechanical risk assessments.

## 8. Conclusions

Nowadays, musculoskeletal disorders continue to be highly prevalent in the working population due to the widespread diffusion of manual activities. Several methods for assessing biomechanical exposure are published in the scientific literature; however, some of these have had scarce application and many lack scientific validation of their predictive claim through longitudinal studies.

Herein, we propose a set of methods provided with a solid scientific background for the assessment of upper-limb exposure to biomechanical risk factors. Some of these tools are, among the many available, those for which more data exist regarding their validity (i.e., hand activity level, ACGIH TLV).

All the described methodologies have been discussed to guide ergonomists and occupational health and safety specialists, who are often required to recommend acceptable workloads and preventive interventions in the workplace.

A tentative classification of the results into categories, intended to advise and guide towards corrective and preventive measures, is also proposed.

Finally, few studies have been performed to address to what extent a given level of biomechanical risk is reduced by means of a shorter work shift duration. To address this shortcoming, we found that the relationship between biomechanical risk factors and work shift duration is approximately linear; more studies are needed to confirm this trend and robustly quantify the risk reduction upon the introduction of part-time work.

## Figures and Tables

**Figure 1 life-15-00109-f001:**
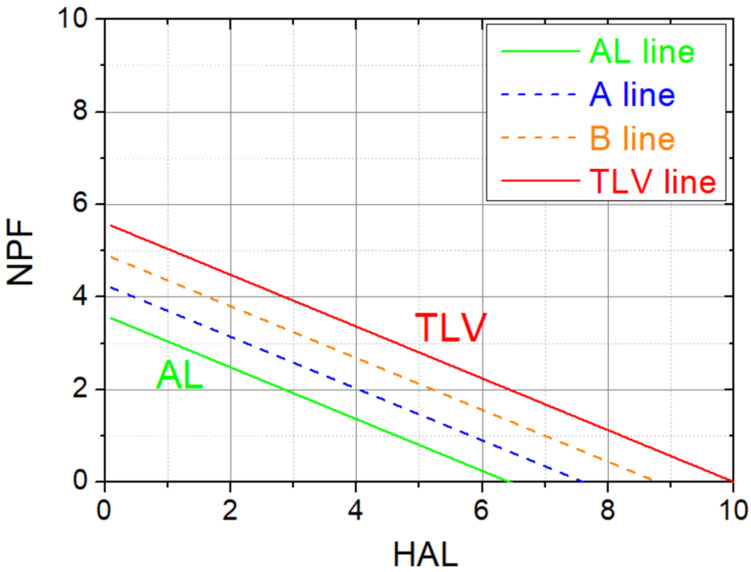
Proposed 5-level classification bands of the ACGIH model, starting from the original AL and TLV lines (in green and red, respectively) with two additional dashed lines, labelled A and B (in blue and orange, respectively).

**Table 1 life-15-00109-t001:** Dictionary of Occupational Titles (DOT) task classification depending on the numerical Lifting Index (LI).

LI (NIOSH)	Task Classification (Dictionary of Occupational Titles/Fitness for Work)
0–1 (Level considered irrelevant)	ML—Very light work Maximum O_2_ consumption: up to 2 MET. MMHv up to 160 min/day: weights of less than 4.5 kg. MMHv over 160 min/day: negligible weights.
1.1–1.5 (First quarter of the interval 1–3)	L—Light Work Maximum O_2_ consumption: 2–3 METs. MMHv up to 160 min/day to: weights less than 9 kg. MMHv over 160 min/day: weights of less than 4.5 kg.
1.6–2 (Second quarter of the interval 1–3)	M—Medium Job Maximum O_2_ consumption: 4–5 METs. MMHv up to 160 min/day: weights between 9 and 23 kg. MMHv over 160 min/day: weights between 4.5 and 11.5 kg.
2.1–2.9 (Upper half of the interval 1–3)	H—Intense work Maximum O_2_ consumption: 6–8 MET. MMHv up to 160 min/day: weights between 23 and 45 kg. MMHv over 160 min/day: weights between 11.5 and 23 kg.
3 or more (Level involving significant biomechanical load)	V—Very intense work Maximum O_2_ consumption: over 8 MET. MMHv up to 160 min/day: weights greater than 45 kg. MMHv over 160 min/day: weights of 23 kg or more.

LI = Lifting Index. NIOSH = National Institute for Occupational Safety and Health. MET = Metabolic Equivalent of Task. MMHv = material handling of vertical loads.

**Table 2 life-15-00109-t002:** Risk category and indications from the BS 8800 task classification depending on the numerical Lifting Index (LI).

LI	Risk Category (Assessment of Tolerability) BS 8800 Table E4	Tolerability: Indications of Necessary Actions and Their Timing BS 8800 Table E5
0–1 (level considered irrelevant)	Very Low (Acceptable)	This level of risk is considered acceptable. No other action is necessary, except the monitoring of the situation (and any prevention and protection measures adopted).
1.1–1.5 (First quarter of the interval 1–3)	Low (Risk to be controlled to be tolerable or acceptable)	No further prevention and protection measures are necessary unless they can be implemented with very limited costs (in terms of time, money, effort). Actions to reduce the level of risk have low priority. It is necessary to monitor the situation (and any prevention and protection measures adopted).
1.6–2 (Second quarter of the interval 1–3)	Medium (Risk to be controlled to be tolerable or acceptable)	Where possible, risk prevention and protection measures should be implemented, down to a tolerable or acceptable level, but the cost should be taken into account. Prevention and protection measures must be implemented within a well-defined period. Monitoring of prevention and protection measures is necessary to ensure they are maintained over time, particularly when the level of risk is associated with harmful consequences.
2.1–2.9 (Upper half of the interval 1–3)	High (Risk to be controlled to be tolerable or acceptable)	Reduce risk. Risk reduction measures should be implemented urgently, indicating a well-defined implementation timeframe. It may be necessary to suspend or limit the activity, or temporarily adopt equivalent risk prevention and protection measures, pending the completion of the reduction measures. The monitoring of prevention and protection measures is necessary to ensure they are maintained over time, particularly when the level of risk is associated with harmful consequences.
3 or more (Level considered at risk for most workers)	Very High (Not acceptable)	A substantial improvement in risk control is needed to reduce it to a tolerable or acceptable level. Work activities should be stopped until risk control measures are implemented and the risk is reduced. If the risk cannot be reduced, work must be inhibited.

LI = Lifting Index.

**Table 3 life-15-00109-t003:** Dictionary of Occupational Titles (DOT) task classification depending on the ACGIH hand activity level.

Hand Activity Level	Task Classification
NPF_ESP_ ≤ NPF_FAL_ (below action level)	ML—Very light work
NPF_FAL_ < NPF_ESP_ ≤ NPF_A_ (between action level and TLV)	L—Light work
NPF_A_ < NPF_ESP_ ≤ NPF_B_ (between action level and TLV)	M—Medium job
NPF_B_ < NPF_ESP_ ≤ NPF_TLV_ (between action level and TLV)	H—Intense work
NPF_ESP_ > NPF_TLV_ (above TLV)	V—Very intense work

NPF = Normalized Peak Force. TLV = threshold limit value.

**Table 4 life-15-00109-t004:** Risk category and indications from the BS 8800 task classification depending on the ACGIH hand activity level.

Manual Activity Level	Risk Category (Assessment of Tolerability) BS 8800 Table E4	Tolerability: Indications of Necessary Actions and Their Timing BS 8800 Table E5
NPF_ESP_ ≤ NPF_AL_ (below action level)	Very low (acceptable)	This level of risk is considered acceptable. No other action is necessary, except the monitoring of the situation (and any prevention and protection measures adopted).
NPF_AL_ < NPF_ESP_ ≤ NPF_A_ (between action level and TLV)	Low (risk to be controlled to be tolerable or acceptable)	No further prevention and protection measures are necessary unless they can be implemented with very limited costs (in terms of time, money, effort). Actions to reduce the level of risk have low priority. It is necessary to monitor the situation (and any prevention and protection measures adopted).
NPF_A_ < NPF_ESP_ ≤ NPF_B_ (between action level and TLV)	Medium (risk to be controlled to be tolerable or acceptable)	Where possible, risk prevention and protection measures should be implemented, down to a tolerable or acceptable level, but the cost should be taken into account. Prevention and protection measures deemed necessary must be implemented within a well-defined period. Monitoring of prevention and protection measures is necessary to ensure that they are maintained over time, particularly when the level of risk is associated with harmful consequences.
NPF_B_ < NPF_ESP_ ≤ NPF_TLV_ (between action level and TLV)	High (risk to be controlled to be tolerable or acceptable)	Reduce risk. Risk reduction measures should be implemented urgently by indicating a well-defined implementation timeframe. It may be necessary to suspend or limit the activity, or temporarily adopt equivalent risk prevention and protection measures, pending the completion of the reduction measures. The monitoring of prevention and protection measures is necessary to ensure that they are maintained over time, particularly when the level of risk is associated with harmful consequences.
NPF_ESP_ > NPF_TLV_ (above TLV)	Very high (not acceptable)	A substantial improvement in risk control is needed so as to reduce it to a level of tolerability or acceptability. Work activities should be stopped until risk control measures are implemented and the risk is reduced. If the risk cannot be reduced, work must be inhibited.

NPF = Normalized Peak Force. TLV = threshold limit value.

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
