# Peer review of "Criteria for Assessing Exposure to Biomechanical Risk Factors: A Research-to-Practice Guide—Part 2: Upper Limbs"

_life, 2025, doi:10.3390/life15010109_

Round 1
Reviewer 1 Report (New Reviewer)
Comments and Suggestions for Authors
Comments on the manuscript of the Life Journal entitled “Criteria for Assessing Exposure to Biomechanical Risk Factors: A Research-to-Practice Guide. Part 2: Upper Limb.”
This manuscript is the second part of the article entitled “Criteria for Assessing Exposure to Biomechanical Risk Factors: A Research-to-Practice Guide—Part 1: General Issues and Manual Material Handling,” recently published in the Life Journal. The subject of this manuscript is related to the assessment of biomechanical risks in the workplace, with a focus on the upper limb. The text is based on the available evidence and a combination of it with the experience and clinical expertise of the authors. It seems that it would have been better to merge the two parts of the article and write a comprehensive guide. I have mentioned my comments regarding this article below.
In the introduction section, it is necessary to express the importance of writing the text in the final paragraphs by logically reviewing the literature and stating the gap.
The purpose of the article should be stated more clearly. Reading the title and introduction of the manuscript, it becomes unclear whether the authors intend to introduce a new method or to review existing methods.
In the introduction, referring to a set of methods for assessing biomechanical risk without providing a clear classification or adequate explanation leads to a lack of proper understanding of the content of the article.
In the “Methods Selected for…” section, briefly mention the type of article, the search method used in scientific databases, the keywords selected, and the data analysis method.
No references are mentioned in some subsections, such as the Hand Activity Level (HAL) and Normalized Peak Force (NPF). Are these sections written entirely based on the experience and knowledge of the authors?
It seems better to use graphs in subsections to better express concepts.
For tables, write a number, title, and description, and refer to the table number in the part of the text that is relevant to that table. Use valid references in tables.
The conclusion section should focus on the results obtained and how they can be applied in the clinic. Some sentences from this section are mentioned in the method section. Please state the key results and suggestions for future studies.
There are systematic review articles and reputable books on upper limb biomechanical risk factors that you can use to improve the quality of your introduction and results section, for example, Biomechanics of the Upper Limbs: Mechanics, Modeling, and Musculoskeletal Injuries.
Author Response
In the introduction section, it is necessary to express the importance of writing the text in the final paragraphs by logically reviewing the literature and stating the gap.
We agree with the Reviewer on the importance of stating the actual gap that our work aims to fill. We felt the need for a practical guide, bridging the gap from scientific research to on-field practice in biomechanical risk assessment. To better highlight this, we added the following sentences:
“Despite the abundance of methods, in the absence of a “gold-standard”, practitioners need for recommendations to approach exposure evaluation on the field. In this work, we present and discuss methods for biomechanical risk assessment of the upper limb, with a solid scientific background and expert consensus, which do not require disproportionate technical, material, financial, and time resources. ”
Our aim was not a comprehensive review of all the biomechanical methods for assessing the upper limb and, indeed, we removed the term “Review” from the title during previous revision runs. We wanted to provide up-to-date methods having more validation data that could be used on-field during a practical risk assessment evaluation. We also introduced oru aim in the Introduction section:
“Our aim is to bridge the gap between research studies and practice, providing also practical recommendations.”
The purpose of the article should be stated more clearly. Reading the title and introduction of the manuscript, it becomes unclear whether the authors intend to introduce a new method or to review existing methods.
The article aim is stated in the last part of the Introduction Section: we want to report and discuss the most validated methods for biomechanical risk assessment, translating their results into risk classification bands for practical evaluations. To better highlight this intent, we added at the end of the Introduction:
“Our aim is to bridge the gap between research studies and practice, providing also practical recommendations.”
In the introduction, referring to a set of methods for assessing biomechanical risk without providing a clear classification or adequate explanation leads to a lack of proper understanding of the content of the article.
The Reviewer is right that we should improve the Introduction section and better explain the content of the article. We modified the sentence as follows:
“Despite the abundance of methods, in the absence of a “gold-standard”, practitioners need for recommendations to approach exposure evaluation on the field. In this work, we present and discuss methods for biomechanical risk assessment of the upper limb, with a solid scientific background and expert consensus, which do not require disproportionate technical, material, financial, and time resources.”
In the “Methods Selected for…” section, briefly mention the type of article, the search method used in scientific databases, the keywords selected, and the data analysis method.
We have not based our research on a systematic review of scientific literature. We tried to be clearer and comprehensive in describing our methods selection at the beginning of Section 2, rephrasing and updating the initial sentence into:
“The selection process was based on both literature review of international scientific papers and expert consensus. Methods with a solid scientific background, that have been published in peer-reviewed journals or recognized books in the occupational field and with a clear description written in the English language are chosen.”
No references are mentioned in some subsections, such as the Hand Activity Level (HAL) and Normalized Peak Force (NPF). Are these sections written entirely based on the experience and knowledge of the authors?
The ACGIH TLV© is based on HAL and PFN, which are measured or assessed, as described by the method itself, on analogic scales from 0 to 10 (where 0=no exposure and 10=the greatest exposure imaginable). HAL and PFN can be combined together in order to compare their combination against ACGIH reference lines and to determine the level of exposure. This approach is clearly described in the ACGIH manual [TLVs and BEIs - ACGIH Portal; 2024] and in different scientific publications, including papers from our research group [Violante FS, Armstrong TJ, Fiorentini C, Graziosi F, Risi A, Venturi S, Curti S, Zanardi F, Cooke RM, Bonfiglioli R, Mattioli S. Carpal tunnel syndrome and manual work: a longitudinal study. J Occup Environ Med. 2007 Nov;49(11):1189-96. doi: 10.1097/JOM.0b013e3181594873; Bonfiglioli R, Mattioli S, Armstrong TJ, Graziosi F, Marinelli F, Farioli A, Violante FS. Validation of the ACGIH TLV for hand activity level in the OCTOPUS cohort: a two-year longitudinal study of carpal tunnel syndrome. Scand J Work Environ Health. 2013 Mar 1;39(2):155-63. doi: 10.5271/sjweh.3312. Epub 2012 Jul 2; Violante FS, Farioli A, Graziosi F, Marinelli F, Curti S, Armstrong TJ, Mattioli S, Bonfiglioli R. Carpal tunnel syndrome and manual work: the OCTOPUS cohort, results of a ten-year longitudinal study. Scand J Work Environ Health. 2016 Jul 1;42(4):280-90. doi: 10.5271/sjweh.3566. Epub 2016 May 6].
We added the reference to the method in the specific subsection.
It seems better to use graphs in subsections to better express concepts.
We agree with the reviewer, and we added a graph to help understanding the 5-level classification of the ACGIH model.
Figure 1: proposed 5-level classification bands of the ACGIH model, starting from the original AL and TLV lines (in green and red, respectively) with two additional dashed lines, labelled A and B (in blue and orange, respectively). The complete equations are reported below.
For tables, write a number, title, and description, and refer to the table number in the part of the text that is relevant to that table. Use valid references in tables.
We thank the Reviewer for the suggestion, and we added the captions and numbered the tables.
The conclusion section should focus on the results obtained and how they can be applied in the clinic. Some sentences from this section are mentioned in the method section. Please state the key results and suggestions for future studies.
Kind Reviewer, we think that our study is more oriented to define practical criteria for assessing exposure to biomechanical risk factors on the field rather than to focus on clinic aspects. However, as suggested, we added a paragraph “Limitations and Future directions” to the manuscript to better emphasize these topics.
“One of the main limitation affecting the methods for risk assessment of manual jobs is related to the estimation of the force. The simultaneous contribution of internal (muscle contraction) and external (the load handled) components poses challenges to objectively evaluate or measure forces actually applied by workers and loads applied on tissues. The use of wearable technologies, which measure pressure/forces (kinetics) and upper arms postures in real-time, while assessing kinematic muscle fatigue, may help to improve this drawback.
However, it has to be noted that the introduction of wearable sensors to monitor manual work in field applications could be complicated, requiring economic resources and skills for signal processing after raw data acquisition. Furthermore, sensors intrusiveness could hamper the feasibility of data recording in real workplaces scenarios (such as in healthcare settings).
Lab-based simulated case studies could be of help, providing benchmark data to compare with in-field biomechanical risk assessment.”
There are systematic review articles and reputable books on upper limb biomechanical risk factors that you can use to improve the quality of your introduction and results section, for example, Biomechanics of the Upper Limbs: Mechanics, Modeling, and Musculoskeletal Injuries.
We thank the Reviewer for the kind suggestion, we tried to improve the Introduction and Results sections following the comments and request that both Reviewers reported. We believe that we have strongly improved the paper quality and readability, better highlighted the scope of our work, and responded to the missing points with clearer explanations and further references. All the addition/revisions are highlighted in yellow in the manuscript for a better evaluation of our modifications.

Reviewer 2 Report (New Reviewer)
Comments and Suggestions for Authors
You have conducted a study on the upper limbs among the criteria for exposure assessment of biomechanical risk factors. The abstract should contain the key contents of the study conducted, divided into an overview, methods, results, and conclusion. However, your study does not meet this requirement. In order to publish the paper in this journal, the following contents should be revised and supplemented.
. Please reorganize the abstract to fit the purpose, method, results, and conclusion of the research.. In the text, please be consistent with the title expressions of 2.1 and 2.2 and 2.3.
. In 3.1, please specify additional titles for each of the two tables, and indicate them in parentheses in the corresponding content of the text.
. In 3.2, please specify additional titles for each of the two tables, and indicate them in parentheses in the corresponding content of the text.
. Please indicate all abbreviations in the table as full terms at the bottom of the table.
Author Response
You have conducted a study on the upper limbs among the criteria for exposure assessment of biomechanical risk factors. The abstract should contain the key contents of the study conducted, divided into an overview, methods, results, and conclusion. However, your study does not meet this requirement. In order to publish the paper in this journal, the following contents should be revised and supplemented.
. Please reorganize the abstract to fit the purpose, method, results, and conclusion of the research.. In the text, please be consistent with the title expressions of 2.1 and 2.2 and 2.3.
A: We thank the Reviewer for the suggestion, we have revised our abstract following the key contents order. However, to be consistent with Part 1 of our work, just published in Life journal, we would avoid the introduction of abstract sub-sections. The complete abstract is now the following:
“Musculoskeletal disorders are the most prevalent occupational health problem and are often related to biomechanical risk factors. Over the last forty years observational methods for exposure assessment have been proposed. To apply them effectively in the field, an in-depth knowledge of each methodology and a solid understanding of their actual predictive value and limitations are required. In this 2-part guide we discuss methods having a solid scientific background, based on expert consensus, and which do not require disproportionate technical, material, financial, and time resources. In Part 1, we focused on the Revised NIOSH Lifting Equation as a validated method for assessing manual material handling and discussed its application when dealing with tasks variability. In Part 2 we looked at methods for the assessment of upper-limb biomechanical exposure in manual jobs. According to the above-mentioned criteria, we discussed methodologies, proposed by the American Conference of Governmental Industrial Hygienists (ACGIH), to evaluate activities requiring high speed continuous movement and use of hand force, working with the arms above the shoulders level, and to prevent localized fatigue in the upper extremities in cyclical work tasks. Finally, a preliminary proposal to proportionate risk assessment to the working duration in part-time jobs is presented.“
. In 3.1, please specify additional titles for each of the two tables, and indicate them in parentheses in the corresponding content of the text.
A: The title for Table 1 and Table 2 were already present; we report them below for the sake of clarity:
“Table 1: Dictionary of Occupational Titles (DOT) task classification depending on the numerical Lifting Index (LI).”
“Table 2: Risk category and indications from the BS 8800 task classification depending on the numerical Lifting Index (LI).”
. In 3.2, please specify additional titles for each of the two tables, and indicate them in parentheses in the corresponding content of the text.
A: The title for Table 3 and Table 4 were already present; we report them below for the sake of clarity:
“Table 3: Dictionary of Occupational Titles (DOT) task classification depending on the ACGIH manual activity level.”
“Table 4: Risk category and indications from the BS 8800 task classification depending on the ACGIH manual activity level.”
. Please indicate all abbreviations in the table as full terms at the bottom of the table.
A: We have added the full terms at the bottom of the Tables as suggested.
Reviewer 3 Report (New Reviewer)
Comments and Suggestions for Authors
Below are some of my comments:
1. 'In Part 1 some general issues relevant to biomechanical risk assessment are discussed, and the method for assessing manual material handling having currently received more robust validation data is reviewed (Revised NIOSH Lifting Equation) together with a discussion about variability of tasks'. Abstract contains much about Part 1 of the article. This can be replaced with further details about this specific study (part 2):
2. Introduction: 'In the first of this two-part guide, general issues regarding occupational biomechanical risk assessment have been discussed...' Please include a citation to the first part.
3. "muscle force, as well as, in the case of the lumbosacral spine, on the forces to which the intervertebral disks are subjected". Please include further details about the amount of force exerted for different manual handling activities.
4. Please also include examples of established thresholds for risk of injury during manual handling injuries.
5. 'different methodologies have been proposed in the scientific literature to assess occupational exposure to biomechanical risk factors. ' Please mention the different methodologies, what are the differences between them. Examples from past studies may be helpful here.
6. Overall, introduction section needs further elaboration. Please also describe human fatigue as well as its types (local/muscle, global) and how it is generally measured in the literature. In general, introduction section needs to contain a description of all concepts in the rest of the paper so that readers unfamiliar with specific jargon can understand.
7. Please provide detailed table captions for all the tables.
8. It may be valuable to include limitations of the current methods, the assumptions made in estimation, errors caused by such assumptions and the potential differences between real-world tasks and lab-based simulated experiments. Please also include potential future directions to further improve the accuracy and reliability of the discussed methods.
Author Response
- 'In Part 1 some general issues relevant to biomechanical risk assessment are discussed, and the method for assessing manual material handling having currently received more robust validation data is reviewed (Revised NIOSH Lifting Equation) together with a discussion about variability of tasks'. Abstract contains much about Part 1 of the article. This can be replaced with further details about this specific study (part 2):
We thank the Reviewer for the comments and revised the abstract, reducing the summary of Part 1 and enhancing the details of our current work (Part 2). The modified abstract is the following:
“In Part 1, the Revised NIOSH Lifting Equation together with a discussion about variability of tasks, are reviewed as more validated methods for assessing manual material handling. Similarly, in Part 2 of this guide we discuss methods proposed by the American Conference of Governmental Industrial Hygienists (ACGIH®) for the assessment of biomechanical exposure and designed to prevent upper limb musculoskeletal disorders as well as localized fatigue. Finally, we introduce a preliminary criterion to proportionate risk assessment to the working duration in part-time jobs.”
- Introduction: 'In the first of this two-part guide, general issues regarding occupational biomechanical risk assessment have been discussed...' Please include a citation to the first part.
The Reviewer is right; we added the reference of our previous paper.
- "muscle force, as well as, in the case of the lumbosacral spine, on the forces to which the intervertebral disks are subjected". Please include further details about the amount of force exerted for different manual handling activities.
The Reviewer is mentioning the Part of our work which has already been accepted and published (Criteria for Assessing Exposure to Biomechanical Risk Factors: A Research-to-Practice Part 1: General Issues and Manual Material Handling). We can’t modify or update that paper, but with that sentence we meant that body posture can influence internal forces. For example, wrist range of motion is strictly related to hand grip and pinch strength (O'Driscoll SW, Horii E, Ness R et. al. The relationship between wrist position, grasp size, and grip strength. J Hand Surg Am 1992; 17: 169-177), while during manual material handling of loads, if the load is moved closer or further from the torso the compressive force at L5/S1 level is greatly affected (pag 137-138 - Chaffin DB, Andersson GBJ, Martin BJ. Occupational biomechanics. 4th Ed. 2006 by John Wiley and Sons).
- Please also include examples of established thresholds for risk of injury during manual handling injuries.
The reference and threshold limits proposed by these methods aim to reduce the occurrence and/or exacerbation of musculoskeletal disorders or diseases. However, when handling task is performed repeatedly above shoulder level, shoulder fatigue may occur increasing the risk of injury. Examples of such working activities include agricultural operations and maintenance activities performed using tools. We added these examples and references for manual material handling (pushing and pulling tasks), changing the end of sub-section 2.2:
“Finally, other manual material handling operations that might resulti in shoulder strain or injury, such as pushing/pulling or lifting/carrying of loads below shoulder height, may coexist with over shoulder work (for example in agricultural operations or maintenance activities). If this is the case, a further lowering of the developed force might be required; threshold values for handling activities performed under the shoulder level may be found in literature.”
- 'different methodologies have been proposed in the scientific literature to assess occupational exposure to biomechanical risk factors. ' Please mention the different methodologies, what are the differences between them. Examples from past studies may be helpful here.
We thank the Reviewer for the comment. We have tried to highlight and stress the value of Takala’s work, (using the bibliographical reference: Takala, E.P.; Pehkonen, I.; Forsman, M.; Hansson, G.Å.; Mathiassen, S.E.; Neumann, W.P.; Sjøgaard, G.; Veiersted, K.B.; Westgaard, R.H.; Winkel, J. Systematic Evaluation of Observational Methods Assessing Biomechanical Exposures at Work. Scand J Work Environ Health 2010, 36, 3–24, doi:10.5271/SJWEH.2876). we also added a sentence in the Introduction to mention the differences between methodologies for assessing occupational exposure and to list some examples regarding available tools.
“Some methods can be considered "general", i.e., such that they provide an assessment of whole-body biomechanical overload to all risk factors (e.g., the Washington State Checklists: https://lni.wa.gov/safety-health/_docs/CautionZoneJobsChecklist.pdf and https://lni.wa.gov/safety-health/_docs/HazardZoneChecklist.pdf). Other tools analyze more in depth specific biomechanical parameters (e.g., the RULA method assesses postural loading in more detail)”.
- Overall, introduction section needs further elaboration. Please also describe human fatigue as well as its types (local/muscle, global) and how it is generally measured in the literature. In general, introduction section needs to contain a description of all concepts in the rest of the paper so that readers unfamiliar with specific jargon can understand.
We thank the Reviewer for the suggestion, we added a paragraph in the Introduction section to describe human fatigue.
“In some occupational settings where workers perform daily manual activities and repeated exertions, muscle fatigue can also occur. Muscle fatigue can be defined as “any reduction in the ability to exert force in response to a voluntary effort”. Fatigue also induce discomfort, pain and, in the long term, can contribute to the development of connective tissue disorders. For this reason, it’s important to predict and limit fatigue in order to preserve workers’, strength, health and work performance. Assessment of localized muscle fatigue could either be assessed objectively (i.e. surface electromyography) , or by rating perceived exertion (RPE).”
- Please provide detailed table captions for all the tables.
We thank the Reviewer for the suggestion and we added the captions and numbered the tables.
- It may be valuable to include limitations of the current methods, the assumptions made in estimation, errors caused by such assumptions and the potential differences between real-world tasks and lab-based simulated experiments. Please also include potential future directions to further improve the accuracy and reliability of the methods discussed.
A: We thank the Reviewer for the suggestion. We added the paragraph “Limitations and Future directions” to the manuscript to take into account these topics as well.
“One of the main limitation affecting the methods for risk assessment of manual jobs is related to the estimation of the force. The simultaneous contribution of internal (muscle contraction) and external (the load handled) components poses challenges to objectively evaluate or measure forces actually applied by workers and loads applied on tissues. The use of wearable technologies, which measure pressure/forces (kinetics) and upper arms postures in real-time, while assessing kinematic muscle fatigue, may help to improve this drawback.
However, it has to be noted that the introduction of wearable sensors to monitor manual work in field applications could be complicated, requiring economic resources and skills for signal processing after raw data acquisition. Furthermore, sensors intrusiveness could hamper the feasibility of data recording in real workplaces scenarios (such as in healthcare settings).
Lab-based simulated case studies could be of help, providing benchmark data to compare with in-field biomechanical risk assessment.”

Round 2
Reviewer 1 Report (New Reviewer)
Comments and Suggestions for Authors
Dear Authors
Hello!
Thank you!
Your answers are quite appropriate and convincing.
Now, I have no more question.
Best
Comments on the Quality of English LanguageIt is better to edit the quality of English language.
Author Response
We thank the Reviewer for the comments and suggestion, that helped us improving the quality of our manuscript. We have checked the text for improving English quality and readability.

Reviewer 2 Report (New Reviewer)
Comments and Suggestions for Authors
Thanks for your efforts.
This manuscript is a resubmission of an earlier submission. The following is a list of the peer review reports and author responses from that submission.
Round 1
Reviewer 1 Report
Comments and Suggestions for Authors
The paper presents the results of a review on criteria for assessing exposure to biomechanical risk factors for upper limb and is a continuation of a previous paper in which some general issues regarding the assessment of biomechanical risk were discussed.
The paper’s main objective was to review the ACGIH method which have more validation data nowadays published for biomechanical risk assessment of the upper limb and postures. The method is focused on the assessment of the risk factors associated with musculoskeletal disorders of the hand, wrist, and forearm and it is aimed to reduce this type of disorders by configuring the conditions to which workers can be exposed without reporting any health issues.
In the Conclusion section, the authors shows that they found that the relationship between biomechanical risk factors and work duration is approximately linear, but they also conclude that more studies are needed to confirm this trend and robustly quantify the risk reduction upon part-time introduction.
The paper is presented in a clear way, is relevant for the field and is well-structured.
The paper contains 4 tables which present the information in a clear way and are easy to interpret and understand.
However, some information should be detailed for more clarity of the paper, such as:
- The abbreviations used in subsection 3.1;
- How the equations in lines 272-275 were established.
The conclusions are consistent with the presented arguments and they respond to the main objective of the study. However, relevant aspects such as the importance of the study, potential beneficiary of the results (researchers, authorities, employers, workers representatives, OHS specialists etc.) and future research directions should be emphasized in the Conclusion section.
The paper contains 31 references which are relevant and appropriate for the scope of the paper. However, the references are older than 5 years and 5 of them are self-citation. Thus, they should be completed with more recent references, published in the last 5 years, and the authors should also consider to reduce the rate of self-citation.
Author Response
The paper presents the results of a review on criteria for assessing exposure to biomechanical risk factors for upper limb and is a continuation of a previous paper in which some general issues regarding the assessment of biomechanical risk were discussed.
The paper’s main objective was to review the ACGIH method which have more validation data nowadays published for biomechanical risk assessment of the upper limb and postures. The method is focused on the assessment of the risk factors associated with musculoskeletal disorders of the hand, wrist, and forearm and it is aimed to reduce this type of disorders by configuring the conditions to which workers can be exposed without reporting any health issues.
In the Conclusion section, the authors shows that they found that the relationship between biomechanical risk factors and work duration is approximately linear, but they also conclude that more studies are needed to confirm this trend and robustly quantify the risk reduction upon part-time introduction.
The paper is presented in a clear way, is relevant for the field and is well-structured.
The paper contains 4 tables which present the information in a clear way and are easy to interpret and understand.
However, some information should be detailed for more clarity of the paper, such as:
- The abbreviations used in subsection 3.1;
A: We explained the abbreviations as suggested by the Reviewer.
- How the equations in lines 272-275 were established.
A: Two of the four equations described in lines 272-275, are the actual TLV and AL lines, reported in the ACGIH manual, namely:
NPFTLV = 5,6-0,56*HAL
NPFAL = 3,6-0,56*HAL
These equations represent two parallel lines with a negative slope equal to -0,56.
To establish two parallel additional lines, thus adapting the ACGIH model into a 5-level classification, we used the same negative slope (-0.56) and mathematically calculated the intercepts: (i) we split the difference between the intercepts of the original ACGIH method (5.6-3.6=2.0) in three equal parts (0.66), then (ii) we added to the intercept of AL line the value 0.66 to find the intercept of the A line (3.6+0.66=4.26), while (iii) added 0.66 to the intercept of the A line to obtain the intercept of the B line (4.26+0.66=4.92). The final equations are the following: NPFA = 4.26-0.56*HAL and NPFB = 4.92-0.56*HAL.
We also improved their description and updated the paragraph as follows:
“To adapt the ACGIH model that provides a classification into three categories to the needs of workload evaluation and risk assessment in 5-level classification bands, two additional Lines (labeled A and B) have been defined (Equations 2 and 3) by di-viding the area between the original AL and TLV Lines (Equations 1 and 4, reported in the ACGIH manual [37] and in [29]), homogeneously. The equations used are as follows:
AL Line: NPFAL = 3,6-0,56*HAL (2)
A Line: NPFA =4 .26-0.56*HAL (3)
B Line: NPFB = 4.92-0.56*HAL (4)
TLV Line: NPFTLV = 5,6-0,56*HAL (5))”
The conclusions are consistent with the presented arguments and they respond to the main objective of the study. However, relevant aspects such as the importance of the study, potential beneficiary of the results (researchers, authorities, employers, workers representatives, OHS specialists etc.) and future research directions should be emphasized in the Conclusion section.
A: We agree with the Reviewer that the Conclusion section needs to be improved: we modified it to highlight the potential beneficiary of these results and what, in our opinion, should the future research directions. The Conclusions now are the following:
“Nowadays musculoskeletal disorders continue to be very prevalent in the working population due also to the widespread diffusion of biomechanical risk factors. Several methods for assessing biomechanical exposure are published in the scientific literature, however, some of these have had scarse application and many lack of scientific validation of their predictive claim through longitudinal studies.
Herein, we propose a set of methods with a solid scientific background for the assessment of the upper limb exposure (distal and proximal part) to biomechanical risk factors. Some of these tools are, amongst the many available, those for which more da-ta exist regarding their validity (i.e. Hand Activity Level ACGIH TLV).
All the described methodologies have been discussed to guide ergonomists and occupational health and safety specialists who are often required to recommend acceptable workloads and preventive interventions in the workplace.
A tentative classification of the results into categories intended to advise and di-rect towards corrective and preventive measures is also proposed.
Finally, few studies have been performed to address to which extent a given level of biomechanical risk is reduced by means of a shorter working schedule. To address this shortcoming, we found that the relationship between biomechanical risk factors and work duration is approximately linear: more studies are needed to confirm this trend and robustly quantify the risk reduction upon part-time introduction.“
The paper contains 31 references which are relevant and appropriate for the scope of the paper. However, the references are older than 5 years and 5 of them are self-citation. Thus, they should be completed with more recent references, published in the last 5 years, and the authors should also consider to reduce the rate of self-citation.
A: We thank the Reviewer for the suggestion comment. We eliminated the oldest of our References to reduce self-citation and similar cited studies. Plus, we added more recent references as suggested throughout the whole manuscript.

Reviewer 2 Report
Comments and Suggestions for Authors
This manuscript appears to build on the authors' previous work in Part-1 (Violante et al., 2024, under revision). However, since the content of Part-1 is unavailable, it is unclear how the two parts are interconnected. Clarification is needed on the division of topics between Part-1 and Part-2 and how they relate rather than functioning as separate studies. Given the long history of research on this topic, the study's background is quite complex, but the authors have not managed this complexity well. The narrative is disjointed and lacks clarity, making the manuscript difficult to follow and reducing its readability, which is a significant shortcoming.
The manuscript title is suggested for minor revision. The title, "Criteria for Assessing Exposure to Biomechanical Risk Factors: A Research-to-Practice Review. Part 2: Upper Limb and Part-Time Work," should be more precise. Both "upper limb work" and "part-time work" are covered in the study, they are treated as separate areas, not "upper limbs’ part-time work."
The term "a research-to-practice review" requires clarification, as its meaning is unclear in the context of the manuscript. Furthermore, only 31 references are cited, which seems insufficient for a review paper. The authors should consider including more literature to support their review.
Although the title mentions both upper limb and part-time work, the discussion on part-time work is minimal (limited to Lines 329-385). This imbalance makes the title seem misaligned with the content.
Sections 1 and 2 (and other sections to a lesser extent) are fragmented. The authors should better integrate their ideas and provide a clearer rationale. In particular, the Introduction section should follow a logical structure: starting with a broad approach, identifying the gap, and gradually introducing the objective of the study. Additionally, it would be helpful to connect the focus on upper limb and part-time work, particularly the latter, more explicitly, possibly in relation to Part 1.
In Line 33, the citation of Waris et al. (1979) may not be suitable for the statement, "In the last 40 years, much work has been performed to characterize the biomechanical risk associated with manual work involving force, speed, and continuity of movements," as the cited paper does not fully support this claim. Moreover, only three references are cited in the Introduction, with the most recent from 2012. These references are inadequate to substantiate the motivation and necessity of the study.
In Lines 21 and 41, "ACGIH®" should be defined or explained for readers unfamiliar with the term, particularly those from non-specialist backgrounds.
The Conclusions section is inappropriate as currently written and requires significant revision. For instance, Lines 391-400 explain the rationale for conducting the study, which belongs in the Introduction, not in the Conclusions.
Author Response
This manuscript appears to build on the authors' previous work in Part-1 (Violante et al., 2024, under revision). However, since the content of Part-1 is unavailable, it is unclear how the two parts are interconnected. Clarification is needed on the division of topics between Part-1 and Part-2 and how they relate rather than functioning as separate studies. Given the long history of research on this topic, the study's background is quite complex, but the authors have not managed this complexity well. The narrative is disjointed and lacks clarity, making the manuscript difficult to follow and reducing its readability, which is a significant shortcoming.
The manuscript title is suggested for minor revision. The title, "Criteria for Assessing Exposure to Biomechanical Risk Factors: A Research-to-Practice Review. Part 2: Upper Limb and Part-Time Work," should be more precise. Both "upper limb work" and "part-time work" are covered in the study, they are treated as separate areas, not "upper limbs’ part-time work."
The term "a research-to-practice review" requires clarification, as its meaning is unclear in the context of the manuscript. Furthermore, only 31 references are cited, which seems insufficient for a review paper. The authors should consider including more literature to support their review.
A: We thank the Reviewer for the suggestion. In order to better clarify that this is not a comprehensive review, we changed the title in:
“Criteria for assessing exposure to biomechanical risk factors: a research-to-practice guide. Part 2: Upper limb”
Similarly, we adjusted some paragraphs in the Introduction section, adding references and improving the narrative flow, as reported:
“Several disorders of the upper limb, which have been linked, with variable degrees of evidence, to manual work involving force, speed, and continuity of movements [4], are still prevalent nowadays [5]: thus, it is extremely important to ensure that workers are not exposed to conditions which may increase the likelihood of development of musculoskeletal upper limb disorders, which are frequent, anyway, among the general population.
Over the years, different methodologies have been proposed in the scientific literature to assess occupational exposure to biomechanical risk factors. [6] In this work, we present and discuss a set of methods for biomechanical risk assessment of the upper limb, with a solid scientific background and that do not require disproportionate technical, material, financial, and time resources to be applied.
In addition, we propose an approach to “translate” the results (often in the form of numerical values / indices) of the biomechanical risk assessment into classification bands to orient ergonomists, occupational health and safety specialists, employers, workers representatives to identify priorities and set up preventive measures.”
Although the title mentions both upper limb and part-time work, the discussion on part-time work is minimal (limited to Lines 329-385). This imbalance makes the title seem misaligned with the content.
A: Thank you very much for your observation. We changed the Title, removing the “part-time”, as reported in the previous answer.
Sections 1 and 2 (and other sections to a lesser extent) are fragmented. The authors should better integrate their ideas and provide a clearer rationale. In particular, the Introduction section should follow a logical structure: starting with a broad approach, identifying the gap, and gradually introducing the objective of the study. Additionally, it would be helpful to connect the focus on upper limb and part-time work, particularly the latter, more explicitly, possibly in relation to Part 1.
A: The Reviewer is right, our Introduction is short due to the presence of Part 1, which better introduces the work. The manuscript was conceived as a single article with the complete evaluation of biomechanical factors, but the Editor suggested to split our study in two manuscripts, thus having more words for better detail and describe the work activity assessment.
We improved the Introduction as explained above, to better clarify the aim of the paper; however, to avoid repetitions/plagiarism we prefer a short introduction, considering that the first part of the manuscript has already been published online. We invite the reviewer to read it for a complete comprehension of the work. https://www.mdpi.com/2075-1729/14/11/1398
We read the whole manuscript and we improved the readability editing Section 2. Together with some minor editing, the main modifications are the following:
- We changed Section 2 title into
“Methods selected for the in-depth, second level analysis of manual activities“
- We added a short paragraph for introducing the Section 2, namely:
“In this section, we present methodologies that can be used for an in-depth evaluation (second level analysis) of manual activities after the preliminary screening (first-level analysis), as described in Part 1 of this guide. The selection process was based on both literature review and expert consensus: methods with a solid scientific background, published in peer-reviewed journals or recognized books in the occupational field, with a clear description written in the English language were first chosen. Then we describe and discuss one method to assess biomechanical overload of the distal part of the upper limb (section 2.1), one tool for the evaluation of the work per-formed above the shoulder level (2.2) and a procedure to define the acceptability of a task in term of upper limb fatigue (2.3).”
- We added:
“For practical purposes, if the overall exposure of workers to repetitive tasks is less than 4 hours, biomechanical analysis can still be carried out according to the ACGIH criteria but the results obtained would lead to an overestimation of the risk. On the other hand, if the overall duration of the repetitive task is less than one hour per day or 5 hours per week, even the ISO 11228-3 standard considers the "repetitiveness" risk factor as negligible: therefore, there would be no need to carry out a specific assessment.”
- We changed a paragraph into:
“Subsequently, many cross-sectional and longitudinal studies have been published validating the method, conducted in different settings, such as manufacturing, healthcare, clerical, and engineering sectors. Recently, longitudinal studies in manufacturing have confirmed the ability of the ACGIH TLV to predict the onset of carpal tunnel syndrome and tendonitis, and it continues to be reviewed and used as a gold standard.”
- We changed the Hand Activity Level description into:
“The Hand Activity Level can be determined by evaluating the average frequency of hand movements during a work-cycle and the duration of the "Duty Cycle" (i.e. the distribution of actual work, where manual effort is greater than 10% of the force exerted in the specific posture, and recovery/rest time).
Alternatively (or in addition) HAL can be estimated by a trained observer who scores the manual task based on an analog scale from 0 (no repetition, hands idle most of the time) to 10 (rapid movements, continuous exertions).”
- We changed the ending of Normalized Peak Force sub-section into:
“where NPF TLV and NPF AL are the peak force values resulting from the intersection of the HAL score (observed) with the TLV and AL lines, respectively. For PFI greater than 1, the respective limit is exceeded.
In multi-tasks jobs, the evaluation can be carried out as a time-weighted average (TWA) according to three different modalities as reported in the ACGIH manual.”
- We changed the beginning of 2.2 sub-section into:
“Given their recent introduction, no field studies are available for these TLVs. However, they are based on a solid scientific background and, being specific for assessing manual tasks performed above shoulder height, they may be useful to design acceptable working conditions. It is thought that most workers can be exposed, day in and day out, below these thresholds without experiencing work-related weariness or shoulder issues. Specifically, the values reported are protective for 75% of women and 95% of men.”
- We changed the ending of 2.2 sub-section into:
“When the force exerted by the hand exceeds the threshold limit of force (TLV), it is recommended to employ suitable control measures to mitigate the risk of shoulder disorders. These measures may include lowering hands height, decreasing hands reach forward, shortening hands force application duration or requiring less force to accomplish the task. It is worth mentioning that also regular over-the-shoulder work, especially if requiring significant force, can cause shoulder fatigue, increasing the risk of discomfort. Thus, to prevent fatigue in repetitive operations ACGIH proposes a specific approach described in the following paragraph (2.3).
Finally, other manual material handling tasks (like pushing or pulling below shoulder height or lifting or carrying loads) are not considered by the TLV and may result in shoulder strain or injury: in these conditions a further lowering of the force might be required.”
- We changed the beginning of 2.3 sub-section into:
“The ACGIH has recommended a TLV to prevent upper limbs fatigue in cyclical work tasks, which has been recently (2022) integrated with further additions. It is thought that most healthy workers can be daily exposed to conditions under the TLV acceptable level. If this is the case, they are expected to maintain their working capacity and normal performances, without undergoing extreme or continuous musculoskeletal fatigue on the whole upper limbs (hands/wrists, forearms, elbows, and shoulders).
The TLV is mainly based on psychophysical data and has been applied in on both field and laboratory studies.”
- We modified the beginning of “Workload Patterns” sub-section into:
“According to ACGIH TLV, workload pattern refers to the ability to repeat and/or sustain biomechanical loads over time. This means that forces and force moments maintain a regular temporal pattern to perform the task (i.e. the same effort required to maintain the worker's weight and the weight of tools, as well as to use them as needed to complete a task.”
- We modified the end of “Workload Patterns” sub-section into:
“A fatigue curve of the TLV is proposed by ACGIH, expressed by the equation of the %MVC in function of the Duty Cycle:
%MVC=100%∙(-0.143∙ln(DC/(100%))+0.066) (1)
where %MVC is the percentage of maximum force or effort of the hand, elbow, or shoulder, and DC is the duty cycle, expressed as the percentage of time during which force is applied compared to the whole activity. Using this fatigue curve/equation, an acceptable duty cycle for a given force (%MVC) or an acceptable %MVC for a certain duty cycle can be mathematically determined.”
And:
“In 2022, ACGIH also made explicit how to calculate the minimum recovery time (RT) from the Duty Cycle (DC) equation (DC=ET/(ET+RT)):
RT=(ET/DC)-ET
in which ET is the effort time and the DC can be derived from the inverse of Equation 0, as a function of MVC.”
In Line 33, the citation of Waris et al. (1979) may not be suitable for the statement, "In the last 40 years, much work has been performed to characterize the biomechanical risk associated with manual work involving force, speed, and continuity of movements," as the cited paper does not fully support this claim. Moreover, only three references are cited in the Introduction, with the most recent from 2012. These references are inadequate to substantiate the motivation and necessity of the study.
A: We thank the Reviewer and updated the references to support this point and enhance the number of references in the paper.
In Lines 21 and 41, "ACGIH®" should be defined or explained for readers unfamiliar with the term, particularly those from non-specialist backgrounds.
A: As you suggested we explained in the text the acronym ACGIH (American Conference of Governmental Industrial Hygienists).
The Conclusions section is inappropriate as currently written and requires significant revision. For instance, Lines 391-400 explain the rationale for conducting the study, which belongs in the Introduction, not in the Conclusions.
A: We thank the Reviewer for helping us improving the Conclusion. In order to comply with both Reviewer’s suggestion, we revised this section that now reads as following:
“Nowadays musculoskeletal disorders continue to be very prevalent in the working population due also to the widespread diffusion of biomechanical risk factors. Several methods for assessing biomechanical exposure are published in the scientific literature, however, some of these have had scarse application and many lack of scientific validation of their predictive claim through longitudinal studies.
Herein, we propose a set of methods with a solid scientific background for the assessment of the upper limb exposure (distal and proximal part) to biomechanical risk factors. Some of these tools are, amongst the many available, those for which more da-ta exist regarding their validity (i.e. Hand Activity Level ACGIH TLV).
All the described methodologies have been discussed to guide ergonomists and occupational health and safety specialists who are often required to recommend acceptable workloads and preventive interventions in the workplace.
A tentative classification of the results into categories intended to advise and direct towards corrective and preventive measures is also proposed.
Finally, few studies have been performed to address to which extent a given level of biomechanical risk is reduced by means of a shorter working schedule. To address this shortcoming, we found that the relationship between biomechanical risk factors and work duration is approximately linear: more studies are needed to confirm this trend and robustly quantify the risk reduction upon part-time introduction.“
